# A Method for Screening Proteases That Can Specifically Hydrolyze the Epitope AA83-105 of α_s1_-Casein Allergen

**DOI:** 10.3390/foods11213322

**Published:** 2022-10-23

**Authors:** Di Liu, Xiaozhe Lv, Yanjun Cong, Linfeng Li

**Affiliations:** 1Beijing Advanced Innovation Center for Food Nutrition and Human Health, College of Food and Health, Beijing Technology and Business University, Beijing 100048, China; 2Department of Dermatology, Beijing Friendship Hospital, Beijing 100050, China

**Keywords:** α_s1_-casein, epitope, mAb, icELISA, protease

## Abstract

Milk protein hydrolysates are common in infant formula, but some of them retain allergenicity due to incomplete hydrolysis of the epitopes for milk allergens. This study explored a method for screening proteases that could specifically hydrolyze the epitope of α_s1_-casein allergen. Firstly, the α_s1_-casein epitope AA83-105 was synthesized by the solid-phase synthesis method. Then, after purification and identification, the complete antigen was prepared through coupling with bovine serum albumin (BSA) and was used to raise monoclonal antibodies (mAb) in BALB/c mice. Additionally, an indirect competitive-enzyme-linked immunosorbent assay (icELISA) was established. The mAb raised against α_s1_-casein protein was used as a control. The results showed that the purity of the synthetic epitope was >90%, and the coupling rate with BSA was 6.31. The mAb subtype is IgG_1_, with a titer of 1:320,000. The mAb reacted specifically with α_s1_-casein but did not cross-react with soybean protein. The linear regression equation of the competitive inhibition curve was y = −9.22x + 100.78 (R^2^ = 0.9891). The detection limit of icELISA method was more sensitive, and the method showed good accuracy and repeatability. The amounts of antigen residues in papain protease hydrolysates were relatively small, and the epitope fragment was detected in papain hydrolysate via mass spectrometry. This study provides ideas and methods for screening proteases that specifically hydrolyze the epitopes of milk allergens and also provides a superior foundation for the development of an advanced hypoallergenic formula.

## 1. Introduction

Food allergy, affecting 6–13% of the global population and of which cow milk allergy is about one-third, is becoming a serious health problem worldwide [1,2]. In China, the latest epidemiological survey suggests that about 2.69% of infants are allergic to milk [3]. In the past decade, the global incidence of milk allergy has been on a rising trajectory, and the reasons remain unclear. Some studies have associated it with the increased usage of infant milk formula based on cow milk proteins, and the most abundant cow milk proteins, such as casein (CN), β-lactoglobulin (β-LG), and α-lactalbumin (α-LA), are all potentially allergenic [4]. An epitope is the part of an allergen protein that is recognized by the immune system, specifically by antibodies, cytotoxic T cells, or B cells.

Currently, the most efficient way to develop hypoallergenic infant formula is to destroy or modify the structure of milk allergens using enzymatic hydrolysis [5]. However, owing to the non-specific hydrolyzation of the epitopes of the proteases, the partially hydrolyzed milk protein hydrolysates used in infant formula still retain allergenicity [6,7]. Antibody-based techniques, such as ELISA, have been extensively developed by researchers to target mostly a single target food allergen, and these ELISA kits were mainly developed based on allergenic protein antibodies [8], which were used to detect the amount of antigen residue in hydrolyzed formula and did not reflect whether the epitope of allergen was hydrolyzed. Therefore, the need to establish a new ELISA method to accurately identify the epitopes of allergens is urgent. α_s1_-casein, an important component of casein, is one of major allergens in cow milk. Previously, our research group randomly synthesized the 15 overlapping peptides of α_s1_-casein and identified the α_s1_-casein IgE epitopes in the sera of patients allergic to cow milk. These peptides included aa6-20, aa11-25, aa21-35, aa26-40, aa91-105, aa126-140, aa141-155, aa171-185, and aa186-200 [9,10].

In present study, we explored and developed a new icELISA method based on one epitope of α_s1_-casein. Considering the epitope of α_s1_-casein at aa91-105 as reported in our past research, this epitope sequence length was extended to allow the allergic region to be fully exposed, the sequence of SEEIVPNSVEQKHIQKEDVPSER (AA83-105) was studied, and the complete antigen was prepared by coupling the epitope SEEIVPNSVEQKHIQKEDVPSER (AA83-105) of α_s1_-casein with BSA. The corresponding mAbs were produced in BALB/c mice. Then, the indirect competitive ELISA method was established to screen the proteases which can hydrolyze the allergenic epitope while the α_s1_-casein protein mAb was used as control.

## 2. Materials and Methods

### 2.1. Preparation, and Identification of Epitope and Complete Antigen

For epitope, the C-terminal amino acids were linked to Wang resin using the Fmoc solid-phase synthesis method, and gradual condensation was performed using the conventional Fmoc method. After the synthesis, the sequence was cut off from the solid-phase carrier using a strong acid, purified by high-performance liquid chromatography (HPLC, Agilent Co., Ltd., Beijing, China), identified by mass spectrometry (MS, Agilent Co., Ltd., Beijing, China), and subjected to freeze-drying for further use [10].

HPLC and MS analysis were performed. HPLC conditions were as follows: chromatographic column: C_18_ column (4.6 mm × 150 mm, 5 μm); mobile-phase A: 0.1% trifluoroacetic acid aqueous solution; mobile-phase B: 80% acetonitrile, 20% water, and 0.09% trifluoroacetic acid; gradient: B increased from 15% to 45% in 0–20 min; flow rate: 1.0 mL/min; detection wavelength: 220 nm; column temperature: 20 °C. MS conditions were as follows [11]: ESI ion source; spray pressure: 15 psi; drying gas temperature: 350 °C; flow rate: 5 L/min; scanning mass range: *m*/*z* 500–2200.

To obtain the complete a_s1_-casein epitope antigen, the purified synthetic peptide was coupled with bovine serum albumin (BSA, Sigma Inc., Shanghai, China) using the glutaraldehyde method, and the coupling ratio was determined via UV scanning (UV spectrophotometer: Japan Hitachi Co., Ltd., Tokyo, Japan) [12].

### 2.2. Preparation, Subtype Identification, Titer Determination, and Specificity of mAb

The complete antigen was used to immunize the BALB/c mice. After four instances of immunization, spleen cells of mice were fused with NS-1 myeloma cells, and the positive hybridoma cells were screened and subcloned multiple times. At optimal cell activity, the hybridoma cells were injected into the abdominal cavity of BALB/c mice, and ascites were collected from the developed abdominal bulge. Meanwhile, mAb against α_s1_-casein were prepared as control. All mice used in this study were cared for in accordance with the Guidelines for the Care and Use of Laboratory Animals published by the U.S. National Institutes of Health (NIH Publication 85-23, 1996), and all experimental studies were conducted under the program approved by the animal ethics committee of Beijing United University (Laboratory Animal Use License No: SYXK(Jing)2017-0038).

Mice IgG_1,_ IgG_2_, IgG_2a_, IgE ELISA kits were used to identify the subtypes according to the manufacturer’s instructions. The titer of mAb was determined via indirect ELISA. The concentration of antigen coating was 5 μg/mL. The used mAb dilutions were 1:5000, 1:10,000, 1:20,000, 1:40,000, 1:80,000, 1:160,000, 1:320,000, and 1:640,000. The negative serum of non-immunized normal mice was used as a negative control.

The specificity of the mAb was determined via western blotting (WB) [13]. Briefly, sodium dodecyl sulfate-polyacrylamide gel electrophoresis (SDS-PAGE) was used to separate α_s1_-casein (Purchasing α-casein that contains 70% α_s1_-casein, Sigma-Aldrich^®^ Inc. Shanghai, China), skim milk, and soy protein isolate. The proteins on the separation gel were transferred to the PVDF membrane (Millipore Co., Ltd., Beijing, China). Then, incubation with mAb corresponding to complete antigen or α_s1_-casein protein was done. Further incubation with the second antibody HRP-sheep anti-mouse IgG (Beijing Friendship Union Co., Ltd., Beijing, China) was carried out. Lastly, 4-chloro-1-naphthol (Sigma Inc., Shanghai, China) was used as the chromogenic substrate solution.

### 2.3. Indirect Competitive ELISA (ic-ELISA)

The protocol of the developed icELISA was as follows: (1)antigen coating: α_s1_-casein was taken as the coating antigen, and carbonate buffer solution (pH 9.6, 50 mmol/L) was used to dilute the α_s1_-casein solution to 10 µg/mL, which was added onto the enzyme-labeled plate (100 µL/well) and kept at 4 °C for 12 h; (2) antigen and primary antibody incubation: adding antigen or sample and a diluted monoclonal antibody at 1:1 ratio to the reaction tube at 4 °C for 12 h and taking the antigen-free or untested sample as a non-competitive reaction system; (3) washing: After coating, the liquid in the wells of the enzyme-labeled plates was removed, and then the plates were washed thrice with 250 µL/well PBST (pH 7.5, 0.02 mol/L phosphate buffer, added 0.1% tween_20_) with vibration lasting 3 min each time. The washing liquid was discarded, and the plate was patted several times on the absorbent paper until there were no obvious drops in the wells; (4) sealing: 150 µL of the sealing liquid (PBST solution containing 1% bovine serum albumin) was added to each well for sealing, which was removed after incubation at 37 °C for 1 h. Later, the plate was washed thrice with PBST (250 µL/well) for 3 min each time. After, the washing liquid was shaken off and patted on the absorbent paper; (5) adding a mixture of primary antibodies and antigens: The monoclonal antibody was added and diluted with antibody diluent (PBST solution containing 1% bovine serum albumin) at a ratio of 1:1000. Then, 100 µL of the antibody was added to each well and incubated at 37 °C for 2 h. (6) After washing, the HRP-sheep anti-mouse IgG_1_ was diluted by 4000 times with antibody diluent. The second antibody (100 µL/well) was added to the wells, covered, and incubated at 37 °C for 1.5 h; (7) after washing, 100 µL/well TMB application liquid (It consists of 10 mL 0.1 mol/L phosphate buffer at pH 6.0, 100 µL 6 mg/mL TMB application solution, and 15 µL 30% hydrogen peroxide) was added to react in the dark at room temperature for 20 min, and the blue color was developed; 2 mol/L of sulfuric acid (50 µL/well) was added to terminate the reaction, with the color changing from blue to yellow, and the absorbance was measured at 450 nm after 30 min.

The icELISA inhibition curve was established based on the optimal working concentration of coated antigen, mAb, and enzyme-labeled second antibody in the present study. α_s1_-casein was diluted in 1280 ng/mL, 640 ng/mL, 320 ng/mL, 160 ng/mL, 80 ng/mL, 40 ng/mL, 20 ng/mL, 10 ng/mL, and 0 ng/mL in order to prepare the standard curve, in which the logarithm of the mass concentration of α_s1_-casein and the competitive inhibition rate were the horizontal and vertical coordinates, respectively. MAb against α_s1_-casein epitope and α_s1_-casein protein were used as the first antibodies, for which the competitive inhibition rate was calculated as follows.
Competition inhibition rate (%) = B/B_0_

B: the OD value of each corresponding concentration of α_s1_-casein competitive inhibition; B_0_: the OD value without α_s1_-casein competitive inhibition.

For detection limit, 10 wells were randomly selected for the detection of zero competitive antigens in indirect competitive ELISA. The average value (D_0_) and standard deviation (SD) of the D_450_ value were calculated. The lower detection limit (LOD) was calculated according to the formula: LOD = (D_0_ − 2SD)/D_0_ × 100% and using the standard curve for the corresponding antigen mass concentration [14]. Additionally, the accuracy and repeatability of the competition inhibition curve were evaluated by the established methods of intra-batch error and inter-batch error.

### 2.4. Preparation of Six Protease Hydrolysates and Determination of Antigen Residues in Hydrolysates

Cow milk α_s1_-casein was used to prepared into 3% (*w*/*v*) aqueous solution. The optimum temperature and pH of protease are according to the conditions given by the manufacturer, as shown in Table 1. The addition amounts of protease were 1500, 2000, 2500, 3000, and 3500 U/g protein. The pH of the reaction system was adjusted with 1 mol/L HCl or NaOH solution to the appropriate pH value of protease and hydrolyzed for 3 h. During the hydrolysis process, the temperature and pH of the solution were kept constant. Immediately after the hydrolysis, the enzyme was inactivated in an 85 °C water bath for 10 min. Six protease hydrolysates were freeze-dried and made ready for use.

The lyophilized powder of different enzymolysis products of α_s1_-casein was added to antigen diluent (50 mmol/L carbonate solution at pH 9.6) to obtain the linear range of the inhibition curve. The antigen residue in hydrolysate was determined using the established indirect competitive ELISA method.

### 2.5. Identification of Hydrolytic Epitope Site of Papain Protease Hydrolysates by MS

The amino acid sequences of six hydrolysate peptides were detected using MS using a Q Exactive HF-X mass spectrometer with a Nanospray Flex (ESI) ion source, ion spray voltage of 2.3 kV, ion transmission tube temperature of 320 °C, data-dependent acquisition mode, mass spectrum scanning range of *m*/*z* = 300−1700, first class mass spectrometer resolution of 70,000, and C-trap maximum capacity of 1 × 10^6^. The maximum injection time of C-trap was 50 ms. The parent ion with top 20 ion strength in the full scan was broken via high energy collision fragmentation (HCD) and detected with secondary mass spectrometry. The resolution of secondary mass spectrometry was 17,500, and the maximum capacity of C-trap was 2 × 10^5^. The maximum injection time of C-trap was 50 ms, the fragment collision energy was 28%, and the threshold intensity was 2.0 × 10^4^. The range of dynamic resistance removal was set to 30 ms.

### 2.6. Determination of Degree of Hydrolysis, Bitterness Value, and Molecular Weight Distribution of Hydrolysate

The determination of hydrolysis degree (DH) of hydrolysate was performed by the OPA method [15]. A standard curve was plotted, taking L-leucine as the standard. 50 mmol/L of the OPA methanol solution (10 mL), 50 mmol/L of N-acetylcysteine-distilled water solution (10 mL), 20% (*w*/*v*) SDS (5 mL), and 0.1 mol/L of borate buffer (75 mL, pH 9.5) were mixed to prepare the OPA reagent. An amount of 3.2 mL of the freshly prepared OPA reagent was taken and placed in a test tube, then added to 400 µL of hydrolysate, i.e., the sample of L-leucine standard, and mixed uniformly. After placing the mixture at 25 °C for 10 min, the absorbance of the sample or standard was measured at 340 nm using an ultraviolet spectrophotometer (Hitachi, Tokyo, Japan). The total content of free amino groups in α_s1_-casein after hydrolysis by hydrochloric acid was determined using an amino acid analyzer (membrapureGmbH, Beijing, China).

DH was calculated with Formula (1)
(1)$$\mathrm{DH}=\frac{\left(NH_{2}\right)T_{x}-\left(NH_{2}\right)T_{o}}{\left(NH_{2}\right)T_{otal}-\left(NH_{2}\right)T_{o}}$$
where (*NH*_2_)*_Tx_* is the content of free amino groups in the hydrolysis product (μmol/mL), (*NH*_2_)*_To_* is the content of free amino groups in unhydrolyzed α_s1_-casein (μmol/mL), and (*NH*_2_)*_Total_* is the content of free amino groups in the α_s1_-casein samples (μmol/mL).

The bitterness value (BV) was determined using the sensory evaluation method [16]. A reference solution was prepared to calibrate the taste intensity of the team members. A series of caffeine (Sigma Inc., Shanghai, China) samples—0 g/L, 3 g/L, 6 g/L, 9 g/L, 12 g/L, and 15 g/L—were prepared, and their BVs were defined as 0, 2, 4, 6, 8, and 10, respectively. After gargling with distilled water, the assessor took about 1.0 mL of enzymatic hydrolysate into their mouth, spat out after 5–10 s, gargled with distilled water, and then evaluated the next sample after 30 s. Ten tasters (all non-smokers) scored samples according to the above benchmark, and the average was used to indicate the degree of bitterness.

The molecular weight distribution of hydrolysates was determined by HPLC (Agilent Co., Ltd., Beijing, China). Chromatographic conditions were as follows: column: TSKgel 2000SWXL 300 mm × 7.8 mm; mobile phase: acetonitrile, water, and trifluoroacetic acid (45:55:0.1 (*v*/*v*/*v*), respectively); detection: UV 220 nm; flow rate: 0.5 mL/min; column temperature: 30 °C; and injection volume: 10.0 μL. The standard samples (relative molecular weight shown in parenthesis) glycine (75 Da) (Sigma Inc., Shanghai, China), Gly-Gly-Gly (189 Da) (Sigma Inc., Shanghai, China), Gly-Gly-Tyr-Arg (451 Da) (Sigma Inc., Shanghai, China), bacitracin (1432 Da) (Sigma Inc., Shanghai, China), Aprotinin (6512 Da) (Sigma Inc., Shanghai, China), and Cytochrome (12,384 Da) (Sigma Inc., Shanghai, China) were used [17].

### 2.7. Application of the Established ic-ELISA Method in Commercial Infant Formula Compared with α_s1_-Casein ELISA Kit

Nestlé infant formula 3 (NF3), Dumex infant formula 2 (DF2), Nestlé infant milk protein partially hydrolyzed formula 1 (NPF1), and Nestlé infant milk protein partially hydrolyzed formula 3 imported from Germany (NPF3) were purchased. These formula powders (dissolved in distilled water) were centrifuged at 4 °C for 15 min at 4800 r/min. The skim milk was removed, freeze-dried, and stored at −20 °C. Then, the skim milk was detected using the established ELISA-method-based onα_s1_-casein epitope AA83-105 and α_s1_-casein ELISA kit (Abcam, Cambridge, UK).

### 2.8. Data Analysis

All the tests of the experiment were carried out in triplicate. The data are presented as mean values ± standard deviations. Statistical evaluation was performed with the student’s *t*-test and analyzed using SPSS 17.0.(Statistical Product Service Solution (SPSS 17.0, IBM Corporation, New York, NY, USA) Statistical significances were detected by one-way analysis of variance (ANOVA), followed by Tukey’s test with an α level of 0.05.

## 3. Results

### 3.1. Preparing of Epitope and Complete Antigen

The synthetic epitope SEEIVPNSVEQKHIQKEDVPSER (aa83-105) was purified using HPLC, and the maximum purity was 92.664%. Meanwhile, the relative molecular weight of the peptide SEEIVPNSVEQKHIQKEDVPSER was 2677.91 Da, as determined by MS, shown in Figure 1 (In order to increase the C-terminal link site of the polypeptide with BSA, an additional lysine is synthesized at the C-terminus, so the molecular weight of the polypeptide determined by mass spectrometry is added to the relative molecular mass of a lysine). The identity of the synthesized peptide was verified. The successful coupling of BSA with synthetic epitopes was performed in the molar molecular ratio (coupling ratio) of 6.31.

### 3.2. Preparation and Identification of mAb

The prepared antibodies were identified according to the instructions of the ELISA *kit*, and their subtypes were identified as IgG_1_. The titers of epitope or α_s1_-casein mAbs were determined by indirect ELISA. The results were summarized in Table 2. P and N represent the OD values of the antibodies and negative serum, respectively, in unimmunized mice. The test results were considered positive when P/N value > 2.1 and negative when P/N value < 2.1. We found that at the antibody dilution 1:320,000, P/N values were >2.1, but at the dilution of 640,000, the optical density (OD) value was <2.1, This suggests that the epitope mAb is effective up to 1:320,000 dilution, and the antibody titer against α_s1_-casein is 1:320,000.

To identify the specificity of the epitope antibody, western blotting was carried out. Because soy proteins are cross-reactive allergens with α_s1_-casein allergen, excluding the reaction between antibodies and soybean protein is an important step in proving the specificity of the epitope antibody. Firstly, SDS-PAGE electrophoresis separation of α_s1_-casein, skim milk, and soy protein were performed, shown in Figure 2A. Lane 3 is skim milk, and the bands, from top to bottom, are lactoferrin, serum albumin, immunoglobulin, α-casein, β-casein, κ-casein, β-lactoglobulin, and α-lactalbumin, respectively. Lane 4 is soybean protein, and the bands are α’subunit, α subunit, β subunit, A3, 11S acid chain, a 38 kDa protein, 11S basic chain, and a 14.4 kDa protein, from top to bottom. The protein bands on the gel were transferred onto the PVDF membrane, and the complete transfer was confirmed via amido black staining. Furthermore, the specificity of mAb against α_s1_-casein epitope and α_s1_-casein protein were determined (Figure 2B–D). The mAb exhibited specific immune reactions against α_s1_-casein epitope and α_s1_-casein protein in skim milk but did not react with soybean protein. Meanwhile, to exclude the false positives, the WB inhibition test was performed with mouse negative serum, which did not react with α_s1_-casein, skim milk, or soy protein.

### 3.3. Establishment of Indirect Competition ELISA

For the first mAb against α_s1_-casein epitope, the standard curve showed a good linear relationship. For the antigen concentration 0–640 ng/mL (Figure 3A), the detection limit was 10.49 ng/mL. Likewise, for the first mAb against α_s1_-casein protein, with the antigen concentration 40–1280 ng/mL, the standard curve showed a good linear relationship (Figure 3B), and the detection limit was 60.50 ng/mL.

In order to prove the accuracy and repeatability of indirect competitive ELISA, intra-batch and inter-batch errors were calculated. Using α_s1_-casein epitope mAb as the first antibody, α_s1_-casein protein was used in five concentration gradients with three parallel experiments and three repeats. The standard deviation was calculated, and the intra-batch error was in the range of 1.159–4.406%, indicating the accuracy of the method. Tests were repeated thrice with different enzyme plates at different times. The standard deviation and coefficient of variation were calculated, and the inter-batch error was expressed by the coefficient of variation between batches, which was in the range of 0.826–4.362%, indicating the reproducibility of the method.

For the mAb of α_s1_-casein protein as the first antibody, the coefficients of variation between wells and batches were 0.828–5.191% and 0.099–4.849% respectively, indicating the good accuracy and reproducibility of the method.

### 3.4. Specific Hydrolysis of α_s1_-Casein Allergenic Epitopes by Various Proteases

The unhydrolyzed α_s1_-casein was diluted as per the detection range of the indirect competitive ELISA inhibition curve. Following the procedure of indirect competitive ELISA, α_s1_-casein epitope mAb and α_s1_-casein protein mAb were used as the first antibody, and the α_s1_-casein contents of sample solution unhydrolyzed by protease were 29.72 ± 5.29 and 30.03 ± 4.02, respectively.

As shown in Figure 4A,B, the antigen residue in the six enzymatic hydrolysates decreased significantly with an increase in enzyme. We observed significant differences in the order of reduction effectiveness of antigen residues in the hydrolysates corresponding to α_s1_-casein epitope mAb and α_s1_-casein protein mAb. For the α_s1_-casein epitope mAb as the first antibody, the antigenic residues were protease M > Neutrase > Protamex > pepsin > Alcalase > papain. Likewise, for the mAb of α_s1_-casein as the first antibody, the antigen residues in the hydrolysate of six different proteases were pepsin > Alcalase > protease M > Protamex > Neutrase > papain.

### 3.5. Identification of Hydrolysis Epitope Sites of Papain Protease Hydrolysates

The fragment EQKHIQKEDVPSERYL releasing from α_s1_-casein epitope AA83-105 (SEEIVPNSVEQKHIQKEDVPSER) was detected in the papain hydrolysate, and no target epitope fragments were detected in other hydrolysates. The mass spectrometry was shown in Figure 5. The results indicated that papain proteases had hydrolyzed the epitope AA83-105 successfully.

### 3.6. Determination of Degree of Hydrolysis, Bitterness Value (BV), and Molecular Weight Distribution of Hydrolysate

The BV of each protease hydrolysate was significant difference using the sensory evaluation method (Figure 6). The pepsin hydrolysate of acid protease and Alcalase was graded with the highest BV of 10.0 and 8.5, respectively. Among the neutral proteases, the BVs of Alcalase, Protamex, and Neutrase hydrolysate were between 7.0–8.0. Notably, the papain hydrolysate was graded with a lower BV of 4.0–5.0, while the protease M hydrolysate had the lowest BV of 1.0–2.0.

The molecular weight distribution of protease hydrolysate was detected by HPLC (Figure 7). Here, the molecular weight distribution of papain hydrolysate is presented, while the others are not shown in order to save space of a whole page. The relative molecular mass correction curve equation was obtained by plotting the retention time (T) of the relative molecular mass logarithm (lgMr) of the standard sample: LgMr = −0.2528T + 7.0984. Based on the standard sample calibration curve, the molecular weight distribution for each enzymolysis was divided into seven parts, in which the proportion of each part represents the peak area to total peak area ratio between the corresponding times (T) of the standard sample. In papain hydrolysate, the proportion of <75 Da increased with an increase in enzyme dosage, ranging from 11.1% to 12.7%. However, the proportion of 75–189 Da was not significantly affected by the enzyme dosage and ranged from 1.4–1.7%. The proportion of 189–451 Da, the largest in the whole hydrolysate, increased significantly with an increase in enzyme dosage (*p* < 0.05), ranging from 37.8–48.7%. On the contrary, the proportions of 451–1432 Da and 1432–6512 Da decreased with an increase in enzyme dosage and ranged from 37.4–33.1% and 10.1–3.8% (*p* < 0.05), respectively. The proportions of 6512–12,384 Da and >12,384 Da were very small and decreased with an increase in enzyme amount, ranging from 1.4–0.21% and 0.23–0.04%, respectively. In a nutshell, with an increase in enzyme dosage, the proportion of components < 451 Da increased, while the proportion of components with >451 Da decreased.

Meanwhile, for each protease hydrolysate, the degree of hydrolysis was significant difference as assessed using the OPA method (Figure 8). The degree of hydrolysis of mild and extensively hydrolysate proteins were <10% and >20%, respectively. The degree of hydrolysis of pepsin in acid protease, considered as mild hydrolysis, was the lowest. Alcalase exhibited a low degree of hydrolysis and hydrolyzed only slightly with the enzyme amount <3500 u. The hydrolysis degree of papain was 10–15% for 1500–3500 u of enzymes. Protease M performed extensively hydrolysis, while Neutrase was stable with about 10% hydrolysis in the 1500–3500 u range. The hydrolysis degree of Protamex was 13% at 3500 u.

### 3.7. Application of the Established ic-ELISA Method in Commercial Infant Formula Compared to α_s1_-Casein ELISA Kit

The residues of α_s1_-casein in four kinds of commercial infant formula were detected using the established ic-ELISA method or α_s1_-casein ELISAkit. The results showed that there were not significant differences for different infant formula (*p* > 0.05) (Figure 9), which indicated that the accuracy of the established ic-ELISA method is similar to that of the α_s1_-casein ELISAkit.

## 4. Discussion

The peptide aa83~105 (SEEIVPNSVEQKHIQKEDVPSER) was identified as the target epitope in this study. The hydrophilicity of the sequence was predicted >0 by the Protean program of DNAStar software, indicating high hydrophilicity. Additionally, the surface accessibility of the sequence being > 1 indicated easy folding and outside exposure. Moreover, the antigen index being > 0 suggested that the sequence is highly likely to form an epitope [18] and confirmed its feasibility. Several epitopes of α_s1_-casein had been identified in previous studies. Although only one epitope was originally prepared with antibodies to establish an indirect competition ELISA method in the present study, the conclusion will provide important ideas for the study of other epitopes of allergens.

BSA and keyhole hemocyanin (KLH) are commonly used coupling carriers for preparing full antigens [19]. BSA is physically and chemically stable, economical, easy to obtain, largely soluble at different pHs and ionic strengths, compatible with some organic solvents, and has no antigenicity. Therefore, it was selected as the coupling carrier in this study. Although some studies showed that BSA was an allergen in cow milk, it did not affect the accuracy of epitope monoclonal antibody results. In the process of preparing monoclonal antibodies, hybridoma cells were screened to specifically react with the epitope without responding to BSA. Thereby, many monoclonal antibodies against the epitope were produced with a specific nature.

The Balb/c mice were immunized with complete antigen of α_s1_-casein epitope, and the mAb was successfully prepared. Meanwhile, the mAb of α_s1_-casein whole protein was prepared as a control. The indirect competitive ELISA method of testing the mAb prepared from the epitope of α_s1_-casein has the obvious advantages of detection limit (<10.49 ng/mL), accuracy, and reproducibility and can be used to detect milk allergens effectively.

Enzymolysis is the mild enzymatic hydrolysis of protein with a specific protease at the optimum temperature and pH to cause less damage to amino acids. Moreover, with the advancement of enzyme technology, novel proteases are enriched and economical. Therefore, enzymatic hydrolysis is the most commonly used method for reducing allergenic protein [20]. Ahmad [21] et al. hydrolyzed buffalo α_s1_-casein with trypsin, α-chymotrypsin, and pepsin at different pHs (pH 2.2 and pH 5.5) and showed with indirect ELISA that the antigenicity was reduced by 85%, 63%, 60%, and 38%, respectively. However, if an epitope is not fully destroyed by protease hydrolysis, the antigenicity remains. Partially hydrolyzed formula (PHF) or extensively hydrolyzed formula (EHF) are well-marketed, but they can still be allergenic due to the presence of allergenic residues [22,23]. However, the present study showed that papain can effectively hydrolyze the epitope aa83~105 of α_s1_-casein. More importantly, this study provides ideas and methods for screening proteases that specifically hydrolyze the epitopes of milk allergens. The hydrolyzed milk protein formula also has advantages. For instance, with the same concentration and amino acid composition, the human body can better absorb these peptides. In particular, the absorption of small peptides (dipeptides and tripeptides) is faster than that of free amino acids [24]. However, formulation of hydrolyzed milk protein infant formula is challenging due to different sources of protein, degrees of hydrolysis, types of proteases, auxiliary processing technologies (such as nonthermal processing), and peptide spectra [25,26]. Therefore, screening protease that can specifically hydrolyze the allergenic epitope is a key necessity [15].

Based on the theoretical hydrolysis sites of the six proteases, excluding pepsin, the other five proteases may hydrolyze the α_s1_-casein epitope aa83-105, however, the conclusion in this original study showed that it is not an effective way to judge the real hydrolysis site for protease because of the complex reaction system of protease and protein substrate. Therefore, establishment of the icELISA method for screening protease via specific hydrolysis of the epitope in α_s1_-casein allergen is necessary and urgent. Our results can provide ideas and effective methods for the specific hydrolysis of food allergen epitopes and set a theoretical basis for the development of non-allergic food.

## 5. Conclusions

An indirect competitive ELISA method using α_s1_-casein epitope mAb was established with high accuracy and repeatability, and the sensitivity of this method is higher than that of the α_s1_-casein allergen method. The method for screening proteases that could specifically hydrolyze the epitope of α_s1_- casein allergen was successfully established. This study provides new ideas for the production of advanced hypoallergenic infant formula.

## Figures and Tables

**Figure 1 foods-11-03322-f001:**
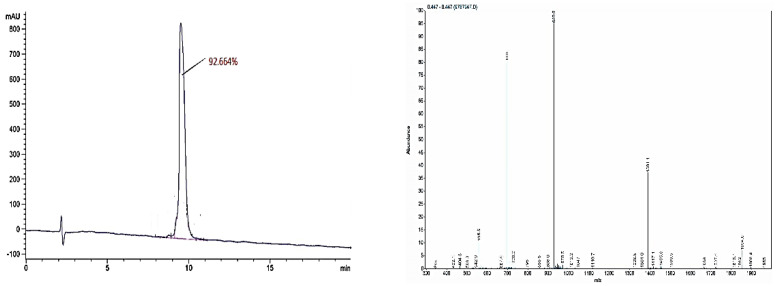
HPLC and MS analysis of synthetic epitope.

**Figure 2 foods-11-03322-f002:**
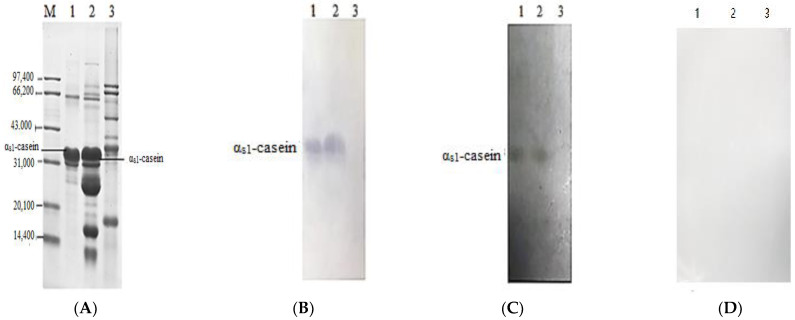
SDS-PAGE electrophoresis profiles (**A**) and immunoblotting of mAb against α_s1_-casein epitope (**B**), α_s1_-casein protein (**C**), or negative serum (**D**). M represents protein markers; Lanes 1, 2, and 3 represent α_s1_-casein, skim milk, and soy protein, respectively.

**Figure 3 foods-11-03322-f003:**
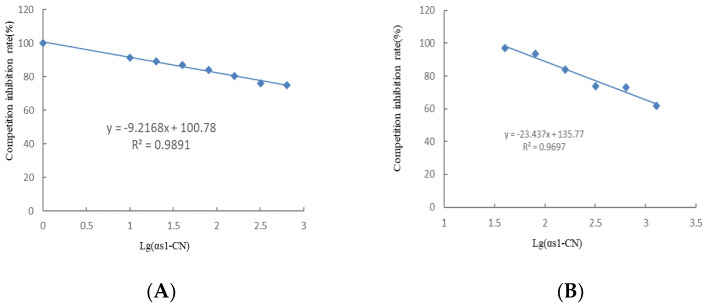
Indirect competitive ELISA inhibition curve of mAb against α_s1_-casein epitope (**A**) and α_s1_-casein protein (**B**).

**Figure 4 foods-11-03322-f004:**
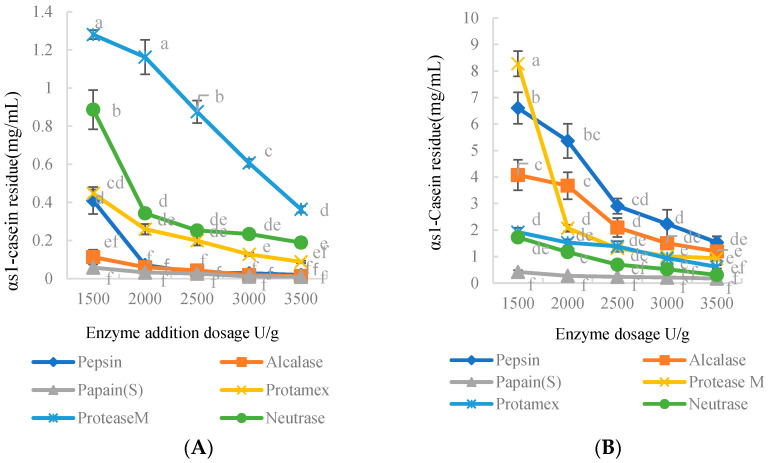
Antigen residues of hydrolysates using mAb against α_s1_-casein epitope (**A**) and α_s1_-casein protein (**B**). a–f indicates the significance of the data differences.

**Figure 5 foods-11-03322-f005:**
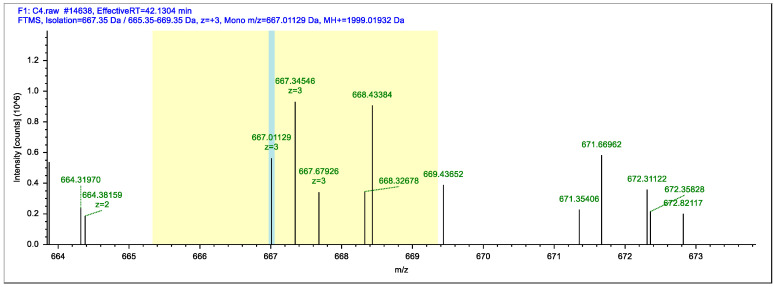
The EQKHIQKEDVPSERYL fragment in papain hydrolysate analyzed via MS.

**Figure 6 foods-11-03322-f006:**
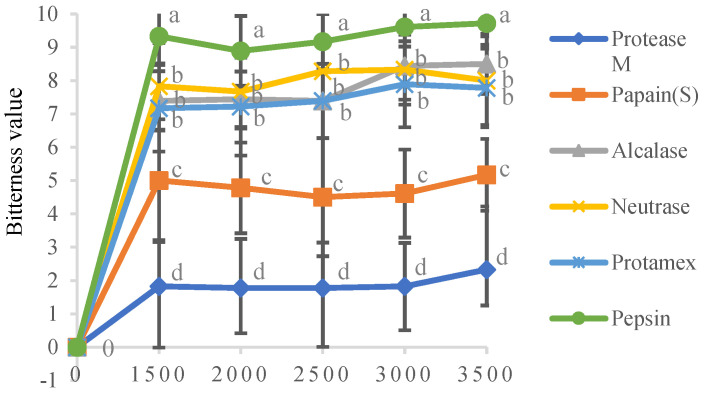
Bitterness values of α_s1_-casein hydrolyzed by six different kinds of proteases. The different letters a–d indicate significant differences between the data.

**Figure 7 foods-11-03322-f007:**
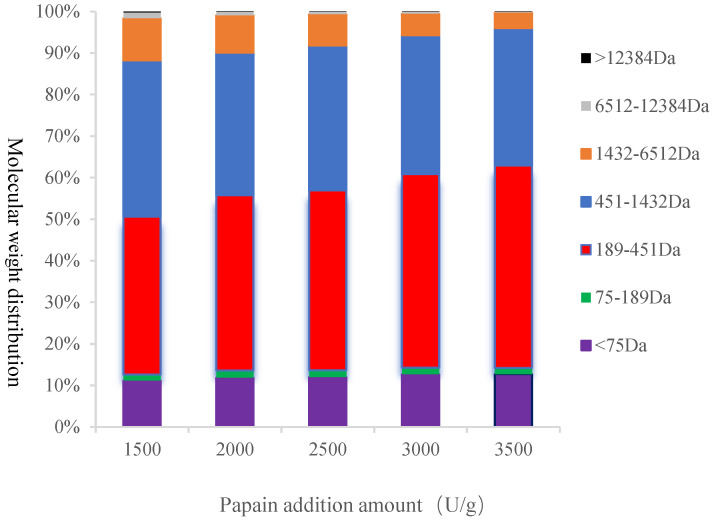
Molecular weight distribution of papain hydrolysate for different enzyme additions.

**Figure 8 foods-11-03322-f008:**
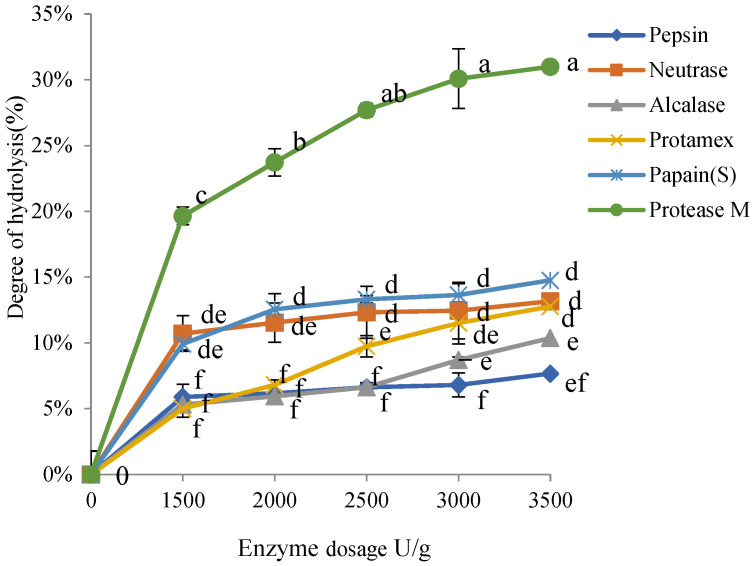
Degree of hydrolysis of α_s1_-casein against six different kinds of proteases. The different letters a–f indicate significant differences between different data.

**Figure 9 foods-11-03322-f009:**
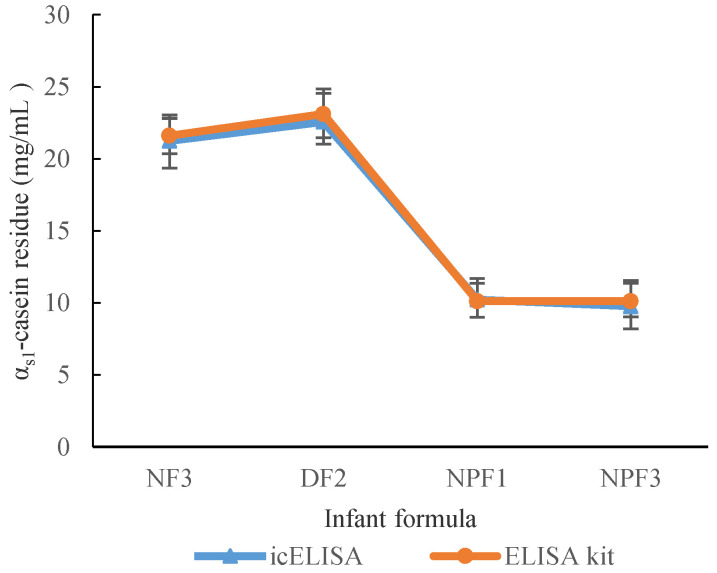
α_s1_-casein residue in commercial infant formula using the established ic-ELISA method or α_s1_-casein ELISAkit.

**Table 1 foods-11-03322-t001:** Optimum conditions for hydrolysis of α_s1_-casein by various proteases.

Protease	Substrate Concentration (%)	pH	Temperature (°C)
Alcalase	3	8.0	60
Protamex	3	7.0	60
Neutrase	3	7.0	50
Papain	3	7.0	60
Pepsin	3	3.0	37
Protease M	3	7.0	50

**Table 2 foods-11-03322-t002:** Absorbance of mAb titer against α_s1_-casein and epitopes.

Dilution Multiple	Epitope mAb (P1)	α_s1_-Casein mAb (P2)	Negative Control (N)
5000	0.505 ± 0.043	0.679 ± 0.034	0.087 ± 0.003
10,000	0.498 ± 0.044	0.632 ± 0.030	0.066 ± 0.002
20,000	0.474 ± 0.051	0.587 ± 0.028	0.054 ± 0.002
40,000	0.427 ± 0.050	0.516 ± 0.030	0.049 ± 0.003
80,000	0.365 ± 0.026	0.411 ± 0.024	0.047 ± 0.002
160,000	0.289 ± 0.027	0.304 ± 0.012	0.046 ± 0.002
320,000	0.225 ± 0.021	0.226 ± 0.024	0.045 ± 0.002
640,000	0.066 ± 0.022	0.057 ± 0.022	0.045 ± 0.002

## Data Availability

The data presented in this study are available on request from the corresponding author.

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
