# Peer review of "A Method for Screening Proteases That Can Specifically Hydrolyze the Epitope AA83-105 of αs1-Casein Allergen"

_foods, 2022, doi:10.3390/foods11213322_

Round 1

Reviewer 1 Report

This is a study showing that the authors established an ELISA using mouse monoclonal antibody against the epitope aa83-105 of as1-casein to compare enzymatic hydrolysis ability of six proteases and found that papain was the most effective protease to hydrolyze the epitope. However, no difference was found in the accuracy to detect the hydrolysis between the new ELISA and commercial ELISA.

There are many points to be brushed up in the presentation.

Major points

1 The authors should describe in the Introduction section why they need a new ELISA to investigate the hydrolysates. As a result, the newly established ELISA has similar ability compared with a commercial ELISA already exist. If the new ELISA has some merit, the title should be changed according to the merit of the new ELISA.

2 Line 49-50: Needs the explanation of as1-casein epitopes and the reason why the epitope (aa83-105) was selected for this investigation. The references of as1-casein epitopes should be listed. The description (Line 319-328) in the Discussion section should be incorporated in this Introduction section.

3 Figure 3: quality of Immunoblotting is rather unclear. Better figure must be presented. 

Minor points

1 Line 51: Reference showing “epitope of BSA” should be described.  

2 L71: “complete antigen” can be “complete as1-casein epitope antigen”

3 L90: Company from which the authors obtained as1-casein protein should be described.

4 L101: description “Mab against epitope and as1-casein” can be “Mab against as1-casein epitope and as1-casein protein” for the better understanding of readers. The description following was also corrected, including Figure 3, Figure 4 and Figure 5.

5 Line 262-264: needs the precise description of the results compared with the results in Figure 1.

6 Line 299: the detail of OPA method should be described in the Materials and Method section.

7 Line 329-332: This sentence is not necessary.

8 Table 1: describe cleaving site for as1-casein protein of these 6 enzymes

9 Table 3: these data should be incorporated in the Figure 6 and Figure 7, respectively.

10 Figure 1: needs explanation of the results.

11 Figure 2 and Figure 3: these 2 figures can be combined.

12 Figure 4: Range of competition inhibition rate is very narrow. Its condition should be improved.

13 Figure 9: no indication of pepsin.

14 Figure 10: It is not easy to distinguish the pattern. Please paint clearer figure.

15 Figure 11: not all proteases were identified.

Author Response

Major points

1 The authors should describe in the Introduction section why they need a new ELISA to investigate the hydrolysates. As a result, the newly established ELISA has similar ability compared with a commercial ELISA already exist. If the new ELISA has some merit, the title should be changed according to the merit of the new ELISA.

Answer: revised

2 Line 49-50: Needs the explanation of as1-casein epitopes and the reason why the epitope (aa83-105) was selected for this investigation. The references of as1-casein epitopes should be listed. The description (Line 319-328) in the Discussion section should be incorporated in this Introduction section.

Answer: revised

3 Figure 3: quality of Immunoblotting is rather unclear. Better figure must be presented. 

 Answer: revised

Minor points

1 Line 51: Reference showing “epitope of BSA” should be described. 

 Answer: Line 51 explained the epitope SEEIVPNSVEQKHIQKEDVPSER of as1-casein was coupled with BSA to prepare the complete as1-casein epitope antigen.

2 L71: “complete antigen” can be “complete as1-casein epitope antigen”

Answer: revised

3 L90: Company from which the authors obtained as1-casein protein should be described.

Answer: From Sigma-Aldrich® Inc., purchasing α-casein that contains 70% αs1-casein.

4 L101: description “Mab against epitope and as1-casein” can be “Mab against as1-casein epitope and as1-casein protein” for the better understanding of readers. The description following was also corrected, including Figure 3, Figure 4 and Figure 5.

Answer: revised

5 Line 262-264: needs the precise description of the results compared with the results in Figure 1.

Answer: revised

6 Line 299: the detail of OPA method should be described in the Materials and Method section.

Answer: revised

7 Line 329-332: This sentence is not necessary.

Answer: reviewer 2 think this sentence is necessary.

8 Table 1: describe cleaving site for as1-casein protein of these 6 enzymes

Answer: The theoretical hydrolysis site of the protease is not the same as the actual hydrolysis site because of complex hydrolysis systems. In this paper, we identified the cleaving site for as1-casein.

9 Table 3: these data should be incorporated in the Figure 6 and Figure 7, respectively.

Answer: We had tried to incorporated these data in Figure 6 and Figure 7 respectively, however, the curves of the protease hydrolysate are unclear because the ordinate values varied greatly. So these data are incorporated in the paragraph.

10 Figure 1: needs explanation of the results.

Answer: In order to increase the C-terminal link site of the polypeptide, an additional lysine is synthesized at the C-terminus, so the molecular weight of the polypeptide determined by mass spectrometry is added to the relative molecular mass of a lysine.

11 Figure 2 and Figure 3: these 2 figures can be combined.

Answer: revised

12 Figure 4: Range of competition inhibition rate is very narrow. Its condition should be improved.

Answer: This is the result of repeated experiments for several times

13 Figure 9: no indication of pepsin.

Answer: revised .Please resize the picture, you can see it.

14 Figure 10: It is not easy to distinguish the pattern. Please paint clearer figure.

Answer: revised

15 Figure 11: not all proteases were identified.

Answer: All proteases were identified. Please resize the picture, you can see them.

Reviewer 2 Report

The authors developed a method for screening proteases that could specifically hydrolyze an epitope of the milk allergen alpha s1 casein. The scientific relevance of this study is high since the final objective is to reduce the allergenicity of hydrolyzed infant formulas, which is of high interest. However, before publication, the authors must carefully revise the manuscript and attend to some important considerations.

The introduction needs to be completely reformulated. The authors barely present the subject and the objective of the paper, without any description of the state of the art of the literature. A brief presentation of milk allergens is required in order to the reader can understand why the authors choose the alphaS1-casein. In the specific case of the alpha S1 casein it is important to refer the reason for the choice of the epitope, since there are several known epitopes in milk allergens. The authors must add information about the existing works studying the hydrolysis of milk allergens as a way to reduce the allergenicity of infant formulas and in what this work will complement the gaps that exists in the literature.

Moreover, as the authors stated in lines 46-48, I do not understand how the formation of bitter peptides is correlated with allergenicity. Why they performed this analysis, since the principal objective is to screen different proteases for the hydrolysis of a specific epitope? This analysis could be performed in a posterior development of an infant formula based on the use of papain for the specifically cleavage of this epitope.

Lastly, the last paragraph must be eliminated since in the introduction it is supposed to mention the objective of the work, but the authors presented here a summary of what they did. A new paragraph must be added including the objective of the work and what the authors intended to do.

The results section is well organized and present objectively the obtained results. The authors produced an impressive number of results, but the discussion about them is quite poor. The authors must improve significantly this section by giving a critical input of the obtained results as well as the relevance of their work for food allergen field. To note that milk allergens present several allergenic epitopes and each individual can react differently to each allergen. This means that this specific epitope could be potentially allergenic for a patient but can also be non-reactive to other patient. Additionally, by eliminating the allergenic potential of one epitope, another epitope can still induce an allergic reaction. This situation must be discussed because it can limit the applicability of this study. Of course, the importance of this study is still high because it improves the knowledge about the specific hydrolysis by proteases but at the same time needs to be complemented with much more information about other epitopes and milk allergens.

Some other minor considerations:

Section 2.3. Describe the protocol of the developed ic-ELISA.

Section 2.6. The citation [14] does not correspond to the description of the method used for sensory evaluation. Please correct or add the description of the method to this section.

Figure 1 has very low resolution. Please add a better image. Add a proper legend to the figure.

Figure 2. Add a proper legend to the figure. Identify each lane with the respective protein. Lines 212-213 can be removed.

Figure 3. Refer in the legend to which code (A, B or C) correspond each image. Lines 217 to 219 can be removed. The immunoblottings need to have a molecular marker to confirm the molecular weights of the obtained bands.

Figure 4 and 5. Place the units of the x axis. Add the error bars to the graphs.

Figures 6, 7 and 9. The statistical analysis must be mentioned and described in the results.

Figure 9. I suggest to use the same colour of each enzyme used in figures 6 and 7.

Still about Figure 9. I guess that the BV values represented in green corresponds to pepsin, but pepsin is not mentioned in the legend. Additionally, there is a mistake on the presentation of the results since the BV values of alcalase are closer to those of protamex and neutrase (7.0-8.0) than to the BV values of pepsin (10.0-8.5), contrarily to what the authors stated in lines 269-274. This fact is also corroborated with the statistical analysis represented in the figure.

Line 278-279. The authors can add these results as supplementary information.

Line 299. How was defined the values for mild and deep hydrolysate degree?

Figure 11. Some colours are missing in the legend of the figure.

The figures need to be formatted. There are several inconsistencies in the axis such as different font style, different colours, lack of units, etc. I think that the figures must be unified by a general formatting/style.

Line 329-332. I did not completely understand this paragraph since the authors stated that BSA has no antigenicity, but BSA is a known milk allergen. I am not quite sure if it is completely correct use BSA as coupling carrier with an allergen such as caseins, which is from the same source (bovine). The authors must clarify this aspect.

The section of conclusion is redundant since this information was already mentioned on the discussion. In my opinion, conclusion can be removed.

Author Response

The authors developed a method for screening proteases that could specifically hydrolyze an epitope of the milk allergen alpha s1 casein. The scientific relevance of this study is high since the final objective is to reduce the allergenicity of hydrolyzed infant formulas, which is of high interest. However, before publication, the authors must carefully revise the manuscript and attend to some important considerations.

The introduction needs to be completely reformulated. The authors barely present the subject and the objective of the paper, without any description of the state of the art of the literature. A brief presentation of milk allergens is required in order to the reader can understand why the authors choose the alphaS1-casein. In the specific case of the alpha S1 casein it is important to refer the reason for the choice of the epitope, since there are several known epitopes in milk allergens. The authors must add information about the existing works studying the hydrolysis of milk allergens as a way to reduce the allergenicity of infant formulas and in what this work will complement the gaps that exists in the literature.

Answer: revised, and the introduction is displayed in red font

Moreover, as the authors stated in lines 46-48, I do not understand how the formation of bitter peptides is correlated with allergenicity. Why they performed this analysis, since the principal objective is to screen different proteases for the hydrolysis of a specific epitope? This analysis could be performed in a posterior development of an infant formula based on the use of papain for the specifically cleavage of this epitope.

Answer: revised

Lastly, the last paragraph must be eliminated since in the introduction it is supposed to mention the objective of the work, but the authors presented here a summary of what they did. A new paragraph must be added including the objective of the work and what the authors intended to do.

Answer: the introduction had been added the objective of the work and what the authors intended to do.

The results section is well organized and present objectively the obtained results. The authors produced an impressive number of results, but the discussion about them is quite poor. The authors must improve significantly this section by giving a critical input of the obtained results as well as the relevance of their work for food allergen field. To note that milk allergens present several allergenic epitopes and each individual can react differently to each allergen. This means that this specific epitope could be potentially allergenic for a patient but can also be non-reactive to other patient. Additionally, by eliminating the allergenic potential of one epitope, another epitope can still induce an allergic reaction. This situation must be discussed because it can limit the applicability of this study. Of course, the importance of this study is still high because it improves the knowledge about the specific hydrolysis by proteases but at the same time needs to be complemented with much more information about other epitopes and milk allergens.

Answer: revised and shown in the discussion section in red font

Some other minor considerations:

Section 2.3. Describe the protocol of the developed ic-ELISA.

Answer: revised

Section 2.6. The citation [14] does not correspond to the description of the method used for sensory evaluation. Please correct or add the description of the method to this section.

Answer: revised and shown in the citation [15]

Figure 1 has very low resolution. Please add a better image. Add a proper legend to the figure.

Answer: revised

Figure 2. Add a proper legend to the figure. Identify each lane with the respective protein. Lines 212-213 can be removed.

Answer: revised

Figure 3. Refer in the legend to which code (A, B or C) correspond each image. Lines 217 to 219 can be removed. The immunoblottings need to have a molecular marker to confirm the molecular weights of the obtained bands.

Answer: revised

Figure 4 and 5. Place the units of the x axis. Add the error bars to the graphs.

Answer: The x-axis is the log value of the antigen concentration, which has no units.

Figures 6, 7 and 9. The statistical analysis must be mentioned and described in the results.

Answer: The statistical analysis had been mentioned and described in the results.

Figure 9. I suggest to use the same colour of each enzyme used in figures 6 and 7.Still about Figure 9. I guess that the BV values represented in green corresponds to pepsin, but pepsin is not mentioned in the legend. Additionally, there is a mistake on the presentation of the results since the BV values of alcalase are closer to those of protamex and neutrase (7.0-8.0) than to the BV values of pepsin (10.0-8.5), contrarily to what the authors stated in lines 269-274. This fact is also corroborated with the statistical analysis represented in the figure.

Answer:The experiments were finished two years ago ,and the raw data can not be found, so we can not change figure 6,7 and 9. Importantly, these data and figures were correct.The BV values of alcalase was added in the results. Please resize the picture, you can see pepsin in the legend.

Line 278-279. The authors can add these results as supplementary information.

Answer: Thank you for your question. To save on layout fees. Thank you

Line 299. How was defined the values for mild and deep hydrolysate degree?

The degree of hydrolysis of mild and extensively hydrolysate proteins were <10% and >20%, respectively.

Figure 11. Some colours are missing in the legend of the figure.

Answer: revised. colours are added.

The figures need to be formatted. There are several inconsistencies in the axis such as different font style, different colours, lack of units, etc. I think that the figures must be unified by a general formatting/style.

Answer: revised.

Line 329-332. I did not completely understand this paragraph since the authors stated that BSA has no antigenicity, but BSA is a known milk allergen. I am not quite sure if it is completely correct use BSA as coupling carrier with an allergen such as caseins, which is from the same source (bovine). The authors must clarify this aspect.

Answer: BSA as coupling carrier had been explained in the discussion.

The section of conclusion is redundant since this information was already mentioned on the discussion. In my opinion, conclusion can be removed.

Answer: The conclusion is the format requirements of the paper. The conclusion had been revised.

Reviewer 3 Report

This manuscript describes the synthesis and purification of an αs2-casein epitope, the production of monoclonal antibodies directed at that epitope, and the development of an indirect ELISA method to detect and quantify the concentration of that epitope.  The method was applied to αs2-casein hydrolysates prepared with several different proteolytic enzymes and papain was discovered to be the protease that was the best choice for hydrolysis of this casein epitope.  The authors indicate that this enzymatic treatment could be used to prepare partial casein hydrolysate for use in hypoallergenic infant formula.

This reviewer has a major concern about the conclusion reached by these authors and their recommendation about the use of this approach for production of infant formula.   Milk has several proteins and several major allergenic proteins – casein, lactoglobulin and lactalbumin.  Casein can be isolated to use as the starting material for production of infant formula.  However, even casein has multiple sub-fractions.  The research described in this manuscript is focused on one epitope in one sub-fraction of casein.  Previous research by these authors and others describes multiple epitopes in the casein sub-fractions and in αs2-casein.  The hydrolysis of only one of these epitopes is unlikely to eliminate the allergenicity of a casein-based hydrolysate formula.  The authors have over-stated the potential future value of this method of formula manufacture.  Furthermore, the authors provide no clinical evidence that the partial hydrolysate developed by this approach actually has reduced allergenicity in milk-allergic human subjects.  Therefore, this key aspect of the study is speculative.

The authors fail to mention that a commercially successful hypoallergenic infant formula based on extensively hydrolyzed casein is available globally.  This casein hydrolysate is made by chemical hydrolysis of casein using strong acid and the resulting product has few peptides.  However, this formula has poor sensory properties.  This situation should be described in the Introduction because it is the main incentive for production of partial hydrolysates.  Commercially successful partial hydrolysates of milk proteins also exist globally.  However, they are not suitable for milk-allergic infants because residual allergenicity remains.  These partial hydrolysates may serve to reduce the likelihood of milk sensitization when fed to infants.  A more complete description of the current hypoallergenic infant formula market and its strengths and weaknesses is needed in the Introduction and Discussion.

This manuscript has a very large number of figures and tables (12 figures, 3 tables).  Table 3 could likely be deleted and the information described in the text.  Figures 4 and 5 could be combined.  So could Figures 6 and 7.

English usage needs improvement also (see specific revisions below):

(1)     Line 20:  “lower” is indefinite and vague; be  more specific

(2)    Line 21:  Change to read “…the amount of antigen residues..”

(3)    Line 24:  Change to read “…and provide a superior…”

(4)    Lines 44-45:  Re-word this sentence “Partially hydrolyzed milk protein as currently used in infant formula retains allergenicity (6,7) due to non-specific and incomplete hydrolysis.”

(5)    Line 90:  Change to read “…protein was done…”

(6)    Line 99:  Change “draw” to “prepare”

(7)    Line 120:  Change to read “…the temperature and pH of the solution was kept constant…”

(8)    Line 121:  Change “killed” to “inactivated”

(9)    Line 166:  Change to read “…r/min, and the skim milk was removed…”

(10)  Change “skimmed” to “skim” throughout

(11) Lines 167-168:  Re-word this sentence “Then the amount of αs2-casein epitope and αs2-casein in the skim milk was determined using the established ELISA methods”

(12) Line 179:  Change “accuracy” to “identity”

(13) Line 234:  Change “testes” to “tests”

(14) Line 277 – 295:  only discuss the results with papain

(15) Lines 299 and 303:  Change “deep” to “extensively hydrolyzed”

(16) Line 312:  Change “significantly” to “significant”

Author Response

This manuscript describes the synthesis and purification of an αs2-casein epitope, the production of monoclonal antibodies directed at that epitope, and the development of an indirect ELISA method to detect and quantify the concentration of that epitope.  The method was applied to αs2-casein hydrolysates prepared with several different proteolytic enzymes and papain was discovered to be the protease that was the best choice for hydrolysis of this casein epitope.  The authors indicate that this enzymatic treatment could be used to prepare partial casein hydrolysate for use in hypoallergenic infant formula.

Answer: This article describes about αs1-casein instead of αs2-casein.

This reviewer has a major concern about the conclusion reached by these authors and their recommendation about the use of this approach for production of infant formula.   Milk has several proteins and several major allergenic proteins – casein, lactoglobulin and lactalbumin.  Casein can be isolated to use as the starting material for production of infant formula.  However, even casein has multiple sub-fractions.  The research described in this manuscript is focused on one epitope in one sub-fraction of casein.  Previous research by these authors and others describes multiple epitopes in the casein sub-fractions and in αs2-casein.  The hydrolysis of only one of these epitopes is unlikely to eliminate the allergenicity of a casein-based hydrolysate formula.  The authors have over-stated the potential future value of this method of formula manufacture.  Furthermore, the authors provide no clinical evidence that the partial hydrolysate developed by this approach actually has reduced allergenicity in milk-allergic human subjects.  Therefore, this key aspect of the study is speculative.

Answer: The relevant content has been added in the introduction and discussion section, shown in the paper for details.

The authors fail to mention that a commercially successful hypoallergenic infant formula based on extensively hydrolyzed casein is available globally.  This casein hydrolysate is made by chemical hydrolysis of casein using strong acid and the resulting product has few peptides.  However, this formula has poor sensory properties.  This situation should be described in the Introduction because it is the main incentive for production of partial hydrolysates.  Commercially successful partial hydrolysates of milk proteins also exist globally.  However, they are not suitable for milk-allergic infants because residual allergenicity remains.  These partial hydrolysates may serve to reduce the likelihood of milk sensitization when fed to infants.  A more complete description of the current hypoallergenic infant formula market and its strengths and weaknesses is needed in the Introduction and Discussion.

Answer: revised, and shown in the paper.

This manuscript has a very large number of figures and tables (12 figures, 3 tables).  Table 3 could likely be deleted and the information described in the text.  Figures 4 and 5 could be combined.  So could Figures 6 and 7.

Answer: Table 3 has been deleted. Figure 4 and Figure 5, or Figures 6 and 7 had been combined.

English usage needs improvement also (see specific revisions below):

(1)     Line 20:  “lower” is indefinite and vague; be  more specific

Answer: revised

(2)    Line 21:  Change to read “…the amount of antigen residues..”

Answer: revised

(3)    Line 24:  Change to read “…and provide a superior…”

Answer: revised

(4)    Lines 44-45:  Re-word this sentence “Partially hydrolyzed milk protein as currently used in infant formula retains allergenicity (6,7) due to non-specific and incomplete hydrolysis.”

Answer: revised

(5)    Line 90:  Change to read “…protein was done…”

Answer: revised

(6)    Line 99:  Change “draw” to “prepare”

Answer: revised

(7)    Line 120:  Change to read “…the temperature and pH of the solution was kept constant…”

Answer: revised

(8)    Line 121:  Change “killed” to “inactivated”

Answer: revised

(9)    Line 166:  Change to read “…r/min, and the skim milk was removed…”

Answer: revised

(10)  Change “skimmed” to “skim” throughout

Answer: revised

(11) Lines 167-168:  Re-word this sentence “Then the amount of αs2-casein epitope and αs2-casein in the skim milk was determined using the established ELISA methods”

Answer: revised. we detected the skim milk using ELISA kit perchased from market and the established ELISA method respectively.

(12) Line 179:  Change “accuracy” to “identity”

Answer: revised

(13) Line 234:  Change “testes” to “tests”

Answer: revised

(14) Line 277 – 295:  only discuss the results with papain

Answer: papain is the final screened protease.

(15) Lines 299 and 303:  Change “deep” to “extensively hydrolyzed”

Answer: revised

(16) Line 312:  Change “significantly” to “significant”

Answer: revised

Round 2

Reviewer 1 Report

None.

Author Response

Thank you very much for your help

Reviewer 2 Report

The authors addressed some suggestions provided by the reviewer. However, other were not completely addressed and in some cases the added information was poor or incomplete. Specifically, the applicability of this method is doubtful if not well supported by the authors. Basic concepts about allergenicity, allergens and epitopes are not mentioned, which it seems to not be well understood by the authors. As mentioned in my previous revision “milk allergens present several allergenic epitopes and each individual can react differently to each allergen. This means that this specific epitope could be potentially allergenic for a patient, but can also be non-reactive to other patient. Additionally, by eliminating the allergenic potential of one epitope, another epitope can still induce an allergic reaction.” These concepts must be added in the discussion section in order to clarify the importance of the work, taking into account that the developed method is only effective for one epitope of one milk allergen.

The format of the submitted manuscript does not conforms to the requirements of “Foods” journal. The submitted manuscript should follow the format of the journal.

Authors should always refer the lines where the corrections were made, because after several corrections they do not correspond to the initial lines. It is very confusing to the reviewers searching for the corrections along the manuscript.

The authors still not mention the state-of-the-art about the works already developed for the hydrolysis of milk allergens as a way to reduce the allergenicity of infant formulas. This information must be added to the introduction since it allows the identification of the gaps that were mentioned by the authors in lines 56-58.

Figure 2. It will be clearer if the authors could substitute the name of each lane by numbers and then make the correspondence to the lane name in figure captions.

Figure 9. The legend for ELISA kit disappeared.

The authors did not clarify the use of BSA as required by this reviewer. This protein is a common milk allergen presenting known antigenicity. Please change accordingly line 373-377.

Reviewer 3 Report

The authors have addressed my major comment and more appropriately discussed the utility of the results presented in this manuscript.  I believe that the manuscript is now sufficiently sound for publication.  Some English language revisions are needed.

Author Response

Thank you very much for your help, your comments are very helpful to me